# Antioxidant and Antifungal Activities and Characterization of Phenolic Compounds Using High-Performance Liquid Chromatography and Mass Spectrometry (HPLC-MS) in *Empetrum rubrum* Vahl ex Willd.

**DOI:** 10.3390/plants13040497

**Published:** 2024-02-09

**Authors:** Carlos Schneider, Makarena González-Reyes, Carola Vergara, Camila Fuica-Carrasco, Patricio Zapata

**Affiliations:** 1Departamento de Ciencias y Tecnología Vegetal, Escuela de Ciencias y Tecnologías, Universidad de Concepción, Campus Los Angeles, Los Angeles 4440000, Chile; m.gonzalezreyes2@uandresbello.edu (M.G.-R.); camilafuica@gmail.com (C.F.-C.); pzapatac1@correo.uss.cl (P.Z.); 2Departamento de Análisis Instrumental, Facultad de Farmacia, Universidad de Concepción, Campus Concepción, Concepción 4030000, Chile; carolavergara@udec.cl

**Keywords:** *Empetrum rubrum*, antioxidant, antifungal activity, *Rhizoctonia solani*, HPLC-MS, polyphenols, anthocyanins, phenolic acids, flavonols

## Abstract

In searching for compounds with antioxidant and antifungal activity, our study focused on the subshrub species *Empetrum rubrum* Vahl ex Willd. (Ericaceae). We measured the antioxidant activity of its methanolic extract (MEE) obtained from the aerial parts (leaves and stems) and of its methanolic extract (MEF) obtained from the lyophilized fruits. The antioxidant activity of the MEE and MEF was evaluated in vitro via a 2,2-Diphenyl-1-picrylhydrazyl (DPPH) free radical and 2,2′-Azino-bis-(3-ethylbenzothiazoline-6-sulfonic acid) diammonium salt (ABTS) cationic radical. The results were expressed in gallic acid and Trolox equivalents for the DPPH and ABTS assays, respectively. The antioxidant activities, for the DPPH and ABTS assays, were also evaluated by considering the IC_50_ values. Concerning the antioxidant activity, the total phenolic content (TPC) in the MEE and MEF was determined using the Folin–Ciocalteu method. Polyphenols contained in the leaves, stems, and fruits of *E. rubrum* were determined qualitatively by employing high-performance liquid chromatography coupled with mass spectrometry (HPLC-MS) analysis. The antifungal activity of the MEE obtained from the aerial parts of *E. rubrum* was tested against *Rhizoctonia solani*. The results of IC_50_ values measured by the DPPH and ABTS methods with MEE were 0.4145 ± 0.0068 mg mL^−1^ and 0.1088 ± 0.0023 mg mL^−1^, respectively, and the IC_50_ values for MEF were 6.4768 ± 0.0218 mg mL^−1^ and 0.7666 ± 0.0089 mg mL^−1^ measured by the DPPH and ABTS methods, respectively. The HPLC-MS analysis revealed the presence of anthocyanins, phenolic acids derivatives, and flavonols. In vitro, mycelial growth of this fungus was reduced from 90% to nearly 100% in the presence of MEE. The observed antifungal effect is related to the presence of the abovementioned phenols, detected in the MEE.

## 1. Introduction

Plants, due to their biodiversity and the broad presence of secondary metabolites [1] in plant tissues, provide a source of antioxidants [2]. A nonenzymatic antioxidant [3] is defined as a molecule capable of stabilizing another that has a missing electron, by giving one [4]. This way prevents reactions with other molecules [5] and prevents oxidative stress [6]. Phenols and polyphenols are intrinsically antioxidants [7], and this group of secondary plant metabolites is present in all parts of plants [8] and possesses nonenzymatic antioxidant properties [9]. Furthermore, plant extracts with phenolic compounds exert different biological functions [10], such as antifungal [11,12], antibacterial [13,14], antiviral [15], and anti-inflammatory activities [16]. Under different environmental biotic stresses, phenols are compounds produced by plants that provide structural integrity and support to plants, and compounds with the aromatic benzene ring play a significant role in plant development, especially in the form of plant defense [17]. Exposure of plants to abiotic stress affects the plants’ secondary metabolism and influences the polyphenolic composition and therefore antioxidant properties [18]. Abiotic stresses are mainly extreme temperature, drought, salinity, and flooding [19], and secondary metabolites such as phenolic compounds can enhance resistance against the effect of abiotic stress [20]. Species of the genus *Empetrum* are keystone species for maintaining mammals and birds and dominate many tundra and heathland ecosystems, excluding other plants through allelopathic toxins [21]. A wide range of biologically active compounds with pharmacological effects are present in the genus *Empetrum*, and *Empetrum* species from the northern hemisphere are used in traditional medicine [22]. Following a common practice and to avoid confusion, all southern hemisphere natural populations are assignable as *E. rubrum* Vahl ex Willd. and all northern hemisphere natural populations as *Empetrum nigrum* L. [21]. Other species of the genus *Empetrum* present in the northern hemisphere are *Empetrum atropurpureum* and *Empetrum eamesii* [23]. There are no scientific reports about biological activities and chemical compounds such as polyphenols in the aerial parts and fruits of *E. rubrum* Vahl ex Willd. This vascular plant lives in high-Andean zones [24] in the presence of living conditions that involve strong winds, high exposition to the sun during the summer months, and low temperatures down to −20 °C. *E. rubrum* is a straight subshrub that develops on sand of volcanic origin and survives after being covered by snow during the winter months [25]. The environmental conditions of *E. rubrum* play a role in the accumulation of polyphenolic compounds that have antioxidant properties, and *E. rubrum* is native to Chile, distributed across the Valparaiso Region to the Magallanes Region in Chile, and in adjacent areas of Argentina [26]. The distribution of *Empetrum nigrum* L. is predominantly circum-arctic-boreal [27], and there are scientific studies about the antioxidant, anti-inflammatory, and α-glucosidase-inhibitory effects of aerial parts’ extract [28], about the antioxidant effect of fruit extracts [29], about antifungal and antibacterial effects [30], and cytotoxic activity against human cancer cells of a compound isolated from the leaves [31] of this species. Considering that the antioxidant and antifungal activities of the southern hemisphere species of the genus *Empetrum* have not been investigated, the present study considers an evaluation of the abovementioned biological activities in methanolic extracts of *E. rubrum*, considering the importance of the identification of phenolic compounds related with antioxidant and antifungal effects. Consequently, to contribute to a preliminary knowledge of secondary metabolites contained in the fruits, stems, and leaves of *E. rubrum*, we also carried out high-performance liquid chromatography (HPLC) and mass spectroscopic (MS) characterization of this species. 

## 2. Materials and Methods

### 2.1. Collection of Wild Plant Material

The leaves, stems, and fruits of *E. rubrum*, commonly known as Brecillo or Murtilla de Magallanes, were collected during the summer from the locality of Callaqui, in the Region of Bio Bio, in Chile, particularly at 1550 m above sea level on the west side of the Callaqui volcano (37°54′57.2″ S 71°28′44.8″ W) on the Andes mountain range. The plant material was transported to the Laboratorio de Extractos Vegetales of the Universidad de Concepción, washed with water, dried at room temperature (25 °C), and stored for its later use. The plant material was authenticated at the Departamento de Botánica of the Universidad de Concepción.

### 2.2. Preparation of the Extracts

To obtain the methanolic extract from the aerial parts (MEE) for the assays mentioned below in points 2.3 to 2.5 and in point 2.7, the dried and ground leaves and stems were treated in a Soxhlet apparatus (Glassco, Dandenong, Australia, 3049/8), carrying out 6 cycles at 65 ± 5 °C and an overnight maceration of 14 h between the cycles. Methanol was used as an extraction solvent for the obtention of the extract. The extract was then concentrated in a rotary evaporator (Heidolph, Schwabach, Germany, Laborota 4001 efficient) under reduced pressure at 65 ± 5 °C. Subsequently, the extract was dried in a glass vessel at 35 °C in an oven and stored at 4 °C until use. The total mass of dry leaves and stems utilized in the three extraction processes was 97.989 g, using a solvent volume of 200 mL in each process. The percentage yield (Y (%)), expressed on the dry weight basis of plant material, was calculated from the following Equation (1):Y (%) = W_1_ × 100/W_2_(1)
where W_1_ is the weight of the extract obtained after solvent removal and W_2_ is the weight of plant material before the extraction procedure.

For the obtention of the methanolic extract from the fruits (MEF), 0.5 g of the lyophilized fruits was dissolved in 25 mL of methanol, and then the mixture was treated with ultrasound (Bransonic, Danbury, CT, USA, Branson 2510) for 30 s, followed by agitation for 16 h. To separate the solvent from the remaining fruits, the mixture was centrifuged (Eppendorf, Hamburg, Germany, Centrifuge 5702), and the process was repeated until the supernatant methanol was colorless, adding pure methanol to the remaining fruits after each centrifugation, carrying out three centrifugation cycles. The supernatant methanol was treated in a rotary evaporator (Heidolph, Schwabach, Germany, Laborota 4001 efficient) and finally was dried in a glass vessel at 35 °C in an oven and stored at 4 °C until use, and the Y (%) was obtained using Equation (1). 

Three kinds of extracts from *E. rubrum* were prepared for the assay mentioned in point 2.6 about HPLC and MS analyses. The first (F1) was obtained by employing distilled water as an extraction solvent from the leaves and stems. The extraction process was performed in a Soxhlet apparatus (Glassco, Dandenong, Australia, 3049/8) at 65 ± 5 °C for two days until the extract surrounding the leaves and stems was colorless. The total mass of dry leaves and stems utilized in the five extraction processes was 92.334 g, using a solvent volume of 200 mL in each process. The drying process for calculating Y (%), calculated with Equation (1), was conducted in the same way as the method for the obtention of MEE, as described above. The second extract (F2) was obtained with methanol from the leaves and stems, which corresponds to the same method for obtaining MEE. The method for obtaining the third extract (F3), from the lyophilized fruits, was the same method employed for the obtention of MEF.

### 2.3. Determination of Antioxidant Activity

#### 2.3.1. Scavenging Activity against DPPH Radicals

The method proposed by Gaviria et al. [32] was used with modifications. The measurements of absorbances were registered with mixtures composed of 2,2-Diphenyl-1-picrylhydrazyl (DPPH) solutions and MEE or MEF. The DPPH solution was prepared by dissolving 1.8 mg of DPPH (Sigma-Aldrich, St. Louis, MO, USA) in methanol in a 50 mL volumetric flask to obtain a concentration of 36 µg mL^−1^, and it was stored at 4 °C protected from light. Before mixing the DPPH solution with MEE or MEF, the DPPH solution was adjusted with methanol to obtain an absorbance of 0.90 ± 0.10 at 517 nm. The mixtures composed of 2.0 mL of DPPH solution and 0.175 mL of different MEE and MEF concentrations were homogenized, and after 1 min of reaction, the absorbance was registered at 517 nm in a spectrophotometer (Merck, Darmstadt, Germany, Spectroquant Pharo 300). The solutions of extracts for this assay were prepared in methanol, in a range of concentrations from 0.04 to 0.80 mg mL^−1^ and in a range of concentrations from 1.00 to 9.00 mg mL^−1^, for MEE and MEF, respectively. The assays were performed in triplicate, and the negative control for all measurements was a mixture of 2.0 mL of DPPH solution and 0.175 mL of methanol. The inhibition percentage of absorbance (IA (%)) was calculated according to Equation (2) as indicated below: IA (%) = (A_0_ − A_1_/A_0_) × 100(2)
where A_0_ is the absorbance of the negative control and A_1_ is the absorbance of the samples. The antioxidant activities performed with the DPPH and 2,2′-Azino-bis-(3-ethylbenzothiazoline-6-sulfonic acid) diammonium salt (ABTS) assays, were calculated based on the inhibitory concentration of MEE and MEF needed to inhibit 50% of the absorbance (IC_50_ value). The IC_50_ values were calculated employing the equation of the curve from the graphs of MEE and MEF concentrations versus IA (%), and those calculated values were employed as the MEE or MEF concentration to carry out the kinetic assays. The kinetic assay consisted of absorbance measurements at 517 nm every 5 min for 3 h, and it was carried out in triplicate. The antioxidant effect of MEE and MEF were also expressed as the gallic acid equivalents (GAE) in the unit of mg of gallic acid per 1000 mg (1 g) of MEE. The results of GAE were obtained from a calibration curve performed in duplicate, and it consisted of DPPH absorbance versus gallic acid (Merck, Hohenbrunn, Germany) concentrations of 0.01, 0.02, 0.03, 0.04, and 0.05 mg mL^−1^. The regression equation of the calibration curves to express the results of GAE for MEE were y = 0.0003x^2^ − 0.0262x + 0.6883 with R^2^ = 0.9875, and y = −0.0108x + 0.6003 with R^2^ = 0.9905 to express the results of GAE for MEF. The GAE was calculated, interpolating the absorbance of a mixture composed of 2.0 mL of DPPH solution and 0.175 mL of MEE or MEF in the calibration curve. The measurements of absorbance at 517 nm were registered after 1 min of reaction. The concentration of 0.175 mL of MEE was 0.500 mg mL^−1^, with 6.272 mg mL^−1^ for MEF. The absorbance measurements of the mixtures mentioned above were performed in triplicate. 

#### 2.3.2. Scavenging Activity against ABTS Radicals

The method developed by Kuskoski et al. [33] was carried out with some modifications. The ABTS radical cation was obtained through the reaction of ABTS (7 mM) (Sigma-Aldrich, St. Louis, MO, USA) with potassium peroxodisulfate (2.45 mM) (Merck, Darmstadt, Germany) in a 50 mL volumetric flask with methanol as the solvent. The reaction was performed at room temperature, for 16 h in darkness. Once the ABTS radical cation was formed, it acquired a dark blue color, and the solution was diluted with methanol to obtain an absorbance of 0.70 ± 0.02 at 734 nm (Merck, Darmstadt, Germany, Spectroquant Pharo 300). After adjusting the ABTS radical cation solution, a 2.0 mL aliquot of this solution was mixed with 0.175 mL of different concentrations of MEE or MEF. After 1 min of reaction, the absorbance was registered at 734 nm. The negative control was a mixture of 2.0 mL of ABTS radical cation solution with 0.175 mL of methanol, and all measurements were performed in triplicate. The inhibition percentage of the absorbance with concentrations of MEE from 0.01 to 0.20 mg mL^−1^, and with concentration of MEF from 0.25 to 3.00 mg mL^−1^, was calculated using Equation (2). With the values of IC_50,_ obtained from the graphs of concentrations of MEE and MEF versus IA (%), a kinetic assay was developed, registering the absorbance values at 734 nm every 5 min for 3 h, and the assays were performed in triplicate. The antioxidant effect was also expressed as Trolox equivalents (TEAC), by interpolating the absorbance of MEE and MEF in the calibration curve, according to the method described by dos Santos et al. [34] with some modifications, and the result was expressed as mg of Trolox per 1000 mg (1 g) of MEE. The absorbance of MEE and MEF was determined in triplicate at 734 nm, with a mixture of 2.0 mL of ABTS radical cation solution and 0.175 mL of MEE or MEF, after one min of reaction. The concentration of MEE and MEF was 0.125 mg mL^−1^ and 0.766 mg mL^−1^, respectively. The calibration curve was performed in duplicate, with ABTS absorbance versus Trolox (Sigma-Aldrich, St. Louis, MO, USA) concentrations of 0.005, 0.010, 0.015, 0.020, 0.025, 0.030, 0.035, 0.040, 0.045, 0.050, and 0.055 mg mL^−1^, and the regression equation of the calibration curve to express the results of TEAC was y = 0.5799 − 0.0143x + 7.2967 × 10^−5^x^2^ with R^2^ = 0.9966 for MEE, and y = 0.6563 − 0.0112x with R^2^ = 0.9905 for MEF.

### 2.4. Assay for Determination of TPC

The determination of the total phenolic content (TPC) in MEE and MEF was carried out using the Folin–Ciocalteu method [35] with modifications, and gallic acid was used as a reference substance for expressing the TPC as gallic acid equivalents in the unit of mg of gallic per 1000 mg (1 g) of MEE or MEF. An aliquot of MEE or MEF, 12.5 mL of distilled water, 1.25 mL of Folin–Ciocalteu reagent (Merck, Darmstadt, Germany), and 5 mL of sodium carbonate (Merck, Darmstadt, Germany) solution (20%), were added in a graduated 25-mL flask, and making up to the mark with distilled water. According to the dilution used, the concentration of MEE in the graduated 25-mL flask was 0.007 mg mL^−1^ and 0.050 mg mL^−1^ for MEF. The absorbance of the homogenized mixture described above was measured at 765 nm (Merck, Darmstadt, Germany, Spectroquant Pharo 300), after thirty min of incubation at room temperature in darkness. The assays were performed in triplicate. The phenol used as a reference for the calibration curve was gallic acid (Merck, Hohenbrunn, Germany) at concentrations of 1, 2, 3, 4, 5, and 6 µg mL^−1^ in two series of six graduated 25-mL flasks because the calibration curve was performed in duplicate. The blank solutions were prepared with all the components, except gallic acid and MEE or MEF, which were not added to the graduated 25-mL flask. The regression equation of the calibration curve to calculate the TPC in MEE was y = 0.041 + 0.1061x + 0.0033x^2^ with R^2^ = 0.9969, and y = 0.0178 − 0.1109x with R^2^ = 0.9955 for calculating the TPC in MEF.

### 2.5. Qualitative Phytochemical Screening of Secondary Metabolites

To detect secondary metabolites, leaves and stems of *E. rubrum* and MEE were subjected to a preliminary phytochemical screening, by utilizing standard procedures for detecting flavonoids, saponins, tannins [36], coumarins, and alkaloids [37]. The results of the qualitative assays were expressed as a marked presence of the metabolites (+++), a normal presence (++), a weak presence of the metabolites (+), and the absence of secondary metabolites was also noted (−). 

### 2.6. HPLC and MS Conditions for the Analyses of Phenolic Compounds

A comparative study of the presence of phenolic compounds in aqueous and methanolic extracts of leaves and stems (F1 and F2, respectively) and fruits (F3) was performed.

Chromatographic analyses of phenolic compounds were conducted with a Shimadzu HPLC NEXERA system (Kyoto, Japan), equipped with a quaternary LC-30AD pump, a DGU-20A5R degasser unit, CTO-20AC column oven, a SIL-30AC autosampler, a CBM-20A controller system, and a UV–Vis diode array (DAD). An SPD-M20A detector was coupled in tandem with a QTrap LC/MS/MS 3200 Applied Biosystems MDS Sciex detector (Foster City, CA, USA). Instrument control and data collection were conducted using CLASS-VP DAD Shimadzu Chromatography Data System and Analyst Software (version 1.5.2).

The anthocyanin analysis in F3 extract was carried out using a C18 column (Kromasil C18 250 × 4.6 mm, 5 μm) with a C-18 pre-column (Nova-Pak Waters, 22 × 3.9 mm, 4 μm) at 40 °C, using a mobile-phase gradient constituted by water/acetonitrile/formic acid (87:3:10% *v*/*v*/*v*) (solvent A) and water/acetonitrile/formic acid (40:50:10% *v*/*v*/*v*) (solvent B). The flow rate was 0.8 mL/min, and the gradient program was from 6 to 30% of solvent B in 15 min, from 3 to 50% in 15 min, from 50 to 60% in 5 min, and from 60 to 6% in 6 min, followed by 9 min of stabilization at 94% [38].

The chromatographic analyses of flavonols and hydroxycinnamic acid derivatives (HCAD) for F1, F2, and F3 extracts were performed according to a method previously described by Ruiz et al. [38] with some modifications. HPLC analyses were carried out on a Kinetex C18 column (core-shell 150 × 4.6 mm, 2.6 μm) with a pre-column (Phenomenex, Torrance, CA, USA) and a binary mobile phase of 0.1% formic acid in water and acetonitrile at a flow rate of 0.5 mL/min, with an injection volume of 10 μL. For flavonols, the mobile-phase gradient ranged from 15 to 25% acetonitrile for 14 min, from 25 to 35% for 11 min, from 35 to 100% for 1 min, and from 100 to 15% for 1 min, with finally a stabilization period of 10 min. The column temperature was set at 40 °C. An additional clean-up step was performed on F3 fruit extract using solid-phase extraction previously described by Ruiz et al. [38], which was used to remove anthocyanins to improve flavonol and HCAD identification. Five mL of the fruit extract was diluted with 5 mL of hydrochloric acid 0.1 N. This solution was loaded on 500 mg Oasis MCX (Waters, Tauton, MA, USA) cartridges previously conditioned with 5.0 mL of methanol and 5.0 mL of water, followed by a rinsing step using 5.0 mL of hydrochloric acid 0.1 N and 5.0 mL of water. The fraction of interest that contained flavonols and hydroxycinnamic acid derivatives was eluted with three 5 mL volumes of methanol. Finally, the pooled solvents were evaporated and resuspended in 5 mL of the mobile phase.

Identity assignment was carried out, considering the retention times and by analysis of DAD and ESI-MS/MS spectra, in positive ionization mode for anthocyanins and negative ionization mode for flavonols and HCAD, under the following parameters: 5 V collision energy, 4000 V ionization voltage, capillary temperature at 450 °C, nebulizer gas 40 psi, and auxiliary gas 50 psi. 

### 2.7. Evaluation of Antifungal Activity of MEE

The antifungal activity of MEE was assayed against the pathogenic fungus *Rhizoctonia solani* Kühn (LBH-Rs-12), obtained from the Laboratorio de Biotecnología de Hongos of the Universidad de Concepción, Campus Los Angeles, and the fungal strain was conserved in potato dextrose agar at 4 °C. 

#### 2.7.1. Evaluation of the Inhibitory Activity of the Fungal Growth

The antifungal activity of MEE, based on the inhibition of the growth of mycelium, was carried out according to Elgorban et al. [39] with some modifications. In sterile Petri dishes, 19 mL of potato dextrose agar (PDA) and 1 mL of MEE concentrations of 100, 200, 300, and 400 mg mL^−1^ were mixed to obtain final MEE concentrations of 5, 10, 15, and 20 mg mL^−1^. Plates with 19 mL of PDA and 1 mL of sterile water were used as the negative control. Mycelial disks of 0.91 cm^2^ obtained from 8-day-old cultures of R. solani, using a sterile cork borer, were inoculated at the center of the plates, and the incubation temperature was 24 °C. The growing area of *R. solani* was measured every 12 h for 3.5 days, using the ImageJ2x software program, and all assays were performed in triplicate. The percentage of inhibitory activity about the area of mycelial growth (IMG (%)) was calculated according to the following Equation (3): IMG (%) = (AC − AT/AC) × 100(3)
where AC is the area of fungal colony without MEE (negative control) and AT is the area of fungal colony treated with different MEE concentrations. 

#### 2.7.2. Evaluation of the Inhibitory Activity of Mycelial Weight

Additionally, to verify the inhibitory activity of MEE against fungal growth, the mycelial biomass was determined by measuring the dry weight. After 3.5 days of incubation as described above, the mycelial dry weight was determined according to Pereira et al. [40] with some modifications. After 4 days of incubation, the plates used as the negative control and the plates with different MEE concentrations were autoclaved at 121 °C and 1 atmosphere for 15 min. After autoclaving, plates containing mycelium were filtered through filter paper, and the filter paper containing the mycelium was dried at 60 °C to a constant weight. Finally, the filter paper containing the dry biomass was weighed, and the dry weight of mycelium was determined using the difference in weight. The assays conducted for determining the percentage inhibition of the dry mycelial weight (IMW (%)) were performed in triplicate, and the following equation was used (4): IMW (%) = (MN − MM/MN) × 100(4)
where MN is the dry mycelial weight without MEE (negative control) and MM is the dry mycelial weight treated with different MEE concentrations.

### 2.8. Statistical Analysis

The results from the absorbance measurements for the DPPH, ABTS, and TPC assays, as well as for the antifungal effect, were obtained using an analysis of variance (ANOVA) and multiple comparisons using the Tukey test (*p* < 0.05) with the statistical software InfoStat (version 2017.1.2). 

## 3. Results and Discussion

### 3.1. Extraction Yield

The total mass of dry leaves and stems was 97.989 g, and the total mass of MEE was 27.739 g. The yield of extraction with methanol was 28.308%, making three extraction processes in all in the Soxhlet apparatus. In each extraction process, the extraction was carried out until the solvent remained transparent and colorless. The total mass for F1 was 18.254 g, extracted with water from 92.334 g of leaves and stems, obtaining a 19.769% yield of extraction after five extraction processes. According to Dias et al. [41], the Soxhlet apparatus was chosen to carry out a conventional technique to determine the total phenolic content and antioxidant capacity, making this technique an effective method for obtaining bioactive compounds. It should be mentioned that the temperature of 65 ± 5 °C employed in the Soxhlet apparatus is related to another study, because in that study the extracts were obtained employing higher temperatures of 80 °C, to evaluate antioxidant activities [42]. 

### 3.2. Antioxidant Activity of MEE and MEF

The results of scavenging activity against DPPH and ABTS radicals reflect the antioxidant activity in all concentrations of MEE and MEF, observing a decrease in IA (%), which is directly proportional to MEE and MEF concentration. The assay performed with DPPH resulted in an IA (%) decrease of 88.83 ± 0.62% and 4.98 ± 0.82% with an MEE concentration of 0.80 mg mL^−1^ and 0.04 mg mL^−1^, respectively. The IA (%) decreases observed in the ABTS assay consisted of 92.63 ± 0.83% and 4.52 ± 0.29% with an MEE concentration of 0.20 mg mL^−1^ and 0.01 mg mL^−1^, respectively. Concerning the antioxidant effect of MEF measured with DPPH, an IA (%) decrease of 73.79 ± 0.41% and 13.86 ± 0.55% was observed with MEF concentrations of 9.00 mg mL^−1^ and 1.00 mg mL^−1^, respectively. The ABTS decrease of IA (%) observed through MEF was 96.69 ± 0.37% and 13.33 ± 0.44% with a MEF concentration of 3.00 mg mL^−1^ and 0.25 mg mL^−1^, respectively. The antioxidant activity reflected in the IC_50_ values of DPPH and ABTS are shown in Table 1, and the GAE and TEAC values are shown in Table 2 and Table 3 for MEE and MEF, respectively. The IC_50_, GAE, and TEAC values are presented as average values (av.) with their standard deviation (SD). In the DPPH assay, gallic acid is commonly used to compare the antioxidant activity of a given substance [43]. On the other hand, in the ABTS assay, as an antioxidant measuring method, Trolox is commonly used as a reference in measurements of antioxidant activity [44]. Considering gallic acid and Trolox as references, we also considered the IC_50_ of both compounds as references to evaluate the IC_50_ obtained in the MEE and MEF assays. The reference values of IC_50_ were calculated from the regression equations of GAE and TEAC mentioned in Section 2.3.1 and Section 2.3.2, and the following values of IC_50_ were obtained: 0.014 mg mL^−1^, and 0.021 mg mL^−1^ of gallic acid for MEE and MEF, respectively, and 0.020 mg mL^−1^ and 0.029 mg mL^−1^ of Trolox for MEE and MEF, respectively. All IC_50_ of the reference compounds have lower values than the IC_50_ values shown in Table 1. For that reason, MEE and MEF have a lower antioxidant capacity than gallic acid and Trolox. It can also be inferred that the higher values of IC_50_, observed for MEF when compared with MEE, are related to lower values of GAE and TEAC observed for MEF.

The antioxidant effect of aerial parts extracted from Korean crowberry (*Empetrum nigrum* var. *japonicum*) has been studied, and a lower value of IC_50_ measured with the DPPH method was reported [28]. However, the method employed to obtain this lower value of 77.99 µg mL^−1^ was different, using a longer reaction time of 30 min between DPPH and the aqueous fraction. The aqueous fraction was obtained from a methanolic extract, and all stages of the extraction process were carried out at room temperature. The method employed by Gao et al. [45], using ultrasound-assisted enzymatic extraction, yielded IC_50_ values of 212.919 µg mL^−1^ and 182.242 µg mL^−1^, measured using the DPPH method and the ABTS method, respectively. The authors noted above performed the extraction from *Empetrum nigrum* aerial parts. Gallic acid was used as a reference substance and as a DPPH scavenger, owing to its effects on health, such as its strong antioxidant and free radical-scavenging activities. Oxidation processes are of significant scientific interest, due to their involvement in the progressive genesis of diseases such as cancer, myocardial infarction, Alzheimer’s dementia, diabetes mellitus, and obesity [46]. On the other hand, Trolox has been used for the ABTS assay, as a reference substance for measuring the antioxidant capacities of plant extracts [47]. The TEAC of MEE and MEF could be compared, with a lower TEAC value of 25.42 ± 1.98 for methanolic extract obtained from *Nassauvia dentata*, and it was collected at the same time as *E. rubrum*, at 1550 m above sea level on the west side of the Callaqui volcano [48]. When assessing the antioxidant effect of MEE at IC_50_ concentration (0.414 mg mL^−1^) for 3 h, a diminution of the percentage of absorbance for DPPH was observed. Just 5 min after that, the antioxidant effect began, a decrease of 73.13% in DPPH absorbance was observed, and subsequently, a recuperation of the free radical was observed (Figure 1).

The recuperation of DPPH was reflected in a gradual decline of the antioxidant effect, reaching 44.62% after 180 min. For the assay with ABTS using IC_50_ concentration (0.108 mg mL^−1^) of MEE to determine the antioxidant effect over 3 h, a decline of 68.17% of absorbance for ABTS was observed in the first 5 min. The decrease in absorbance for ABTS showed an increase until min 100, reaching a decrease of 91.11%. Subsequently, the antioxidant effect of MEE remained practically constant, with slight variations of 1–2% (Figure 1). 

About the antioxidant effect of MEF over 3 h, an evident diminution of up to 80% approximately in the first 10 min was produced by MEF at IC_50_ concentrations on the absorbances of DPPH and ABTS (Figure 2). After ten min of reaction, an increase in the antioxidant effect over the time, due to MEF, in the presence of DPPH and ABTS was observed, reaching a diminution of 94.54% and 95.27% at 180 min for DPPH and ABTS, respectively. 

### 3.3. Determination of TPC

According to the measurements carried out in triplicate and based on the regression equation of the calibration curve, 0.007 mg mL^−1^ of MEE is equivalent to 1.683 ± 0.024 µg mL^−1^ of gallic acid. Consequently, the TPC expressed as gallic acid equivalents is 240.428 ± 3.428 mg of gallic per 1000 mg (1 g) of MEE. This result for MEE, in comparison with the total phenols from *Empetrum nigrum* aerial parts, extracted with ethanol at concentrations of 50 to 70% and made under different extraction conditions [49], revealed a higher content of phenols in MEE. In the same way, the TPC in MEF was calculated, considering the regression equation, to obtain an equivalence of 2.184 ± 0.045 µg mL^−1^ of gallic acid in 0.050 mg mL^−1^ of MEF. Therefore, the TPC expressed as gallic acid equivalents is 43.680 ± 0.901 mg of gallic per 1000 mg (1 g) of MEF. 

### 3.4. General Qualitative Analysis of Secondary Metabolites

According to the phytochemical screening (Table 4), leaves, stems and MEE contain a marked presence of flavonoids and condensed tannins, and these classes of secondary metabolites are directly related to the observed antioxidant effects of MEE [50]. 

### 3.5. HPLC and MS Analyses of Phenolic Compounds

The anthocyanin chromatographic profile for *E. rubrum* fruit extract F3 shows (Figure 3, Table 5) that the main anthocyanins identified are delphinidin-3-glucoside and malvidin-3-glucoside, followed by cyanidin and petunidin-3-glucosides. In a lesser proportion, some pentoside conjugates can be observed, but their identity was not confirmed with commercial standards, as the glucoside derivatives were. Other authors [51] identified the anthocyanin profile in other *Empetrum* berries (Black Crowberry) with the main identified anthocyanins being delphinidin, cyanidin, peonidin, and malvidin, but conjugated with galactose instead of glucose.

The phenolic compounds chromatographic analysis of the F1 (leaves and stems aqueous extract), F2 (leaves and stems methanolic extract), and the cleaned F3 fruit extract showed that the main compounds were HCAD and conjugated flavonols (Figure 4). There was no significant presence of flavanol or larger-molecular-weight procyanidins. The principal flavonols found in the samples were quercetin-3-glucoside and a laricitin hexoside, which can be either galactoside or glucoside, in the stem and leaves samples. The other significant signals were tentatively identified as a caffeoylquinic acid isomer and a coumaric acid and caffeic acid derivate (Table 6), also more prevalent in stem and leaves samples. The F3 sample, which was previously treated to remove the anthocyanins, maintained the same proportion of flavonols and HCAD, but in much lesser quantities. Laaksonen et al. [52] analyzed flavonol and HCAD in *Empetrum nigrum* berries and found that the main flavonols identified were similar in profile with galactosides of laricitin and quercetin and coumaric acid derivates.

### 3.6. Antifungal Activity of MEE

The observed antifungal activity of MEE against *R. solani* consisted of a growth inhibition of mycelium (Figure 5) and a diminution of the mass of the fungus. The reduction in both parameters mentioned above was directly related to the MEE concentration. As shown in Figure 6, the maximum concentration of 20 mg mL^−1^ inhibited the mycelial growth in a percentage of 97.70 ± 0.72% after 84 h of cultivation. 

For the inhibition of the mass of fungus, and as seen in Figure 7, the same concentration of 20 mg mL^−1^ mentioned above leads to a decrease of 98.10 ± 1.34% after 84 h.

*R. solani* was chosen because it causes legume yield loss all over the world. The loss due to *R. solani* has been estimated at an average of 20%, and even, in rare scenarios, 30–60%, and a complete loss of the legume crops has also been observed [53]. Furthermore, *R. solani* in Chile is an important and recurrent soil phytopathogen, affecting the quality and yields of plants [54].

Similar to our results, the research of an algal methanolic extract performed by Al-Nazwani et al. [55] demonstrated that there is a correlation between the increase in the extract concentration tested, and the inhibition of growth and the biomass decline of *R. solani*. The authors mentioned above also concluded that phenol, in comparison with other tested compounds, exerted the greatest antifungal effect, reflected in the mycelial growth of *R. solani*. Other research demonstrated the association of the antifungal and antioxidant activities of methanolic plant extracts with their phenols, tannins, and flavonoids content [56]. According to the observed inhibition of mycelial growth of *R. solani*, *Sclerotium rolfsii*, and another fungus, a higher content of total phenols and flavonoids in a 70% ethanolic extract, in comparison with an aqueous extract, is directly related to the growth inhibition of these microorganisms [57]. The research mentioned above agrees with the higher levels of TPC detected in MEE, in comparison with TPC detected in the methanolic extract obtained from *Nassauvia dentata* [48], because MEE has a higher antifungal effect than the methanolic extract from *Nassauvia dentata*, against *R. solani.* Another modality of antifungal activity consists of biocontrol agents against the fungus *Botrytis cinerea*, using microorganisms inhabiting the leaf surface of *E. rubrum* [58].

## 4. Conclusions

The results of our research revealed a considerable content of phenols extracted from the aerial parts of *E. rubrum*, using a polar solvent such as methanol, and these phenol compounds exhibited antioxidant activity and antifungal activity. Furthermore, anthocyanins, phenolic acid derivatives, and flavonols of *E. rubrum* are related to an antifungal effect against *R. solani.* Concerning the observed antifungal effect, extracts obtained with polar solvents from *E. rubrum* could be assayed against other phytopathogen fungi, and the methanol extract obtained from *E. rubrum* may be a natural antifungal agent in the future.

## Figures and Tables

**Figure 1 plants-13-00497-f001:**
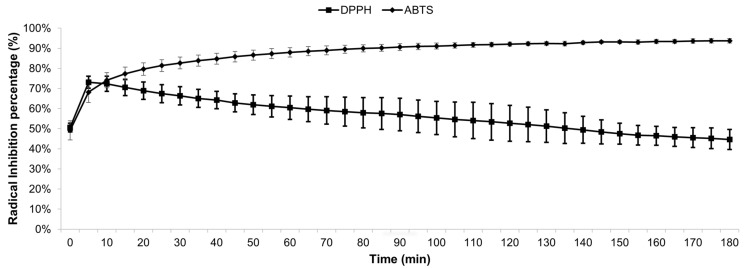
Antioxidant effect of MEE over 3 h, using IC_50_ concentrations.

**Figure 2 plants-13-00497-f002:**
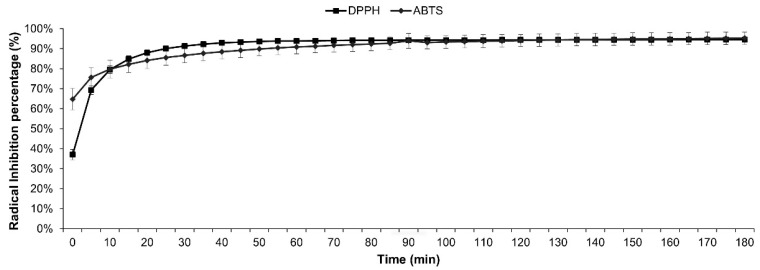
Antioxidant effect of MEF over 3 h, using IC_50_ concentrations.

**Figure 3 plants-13-00497-f003:**
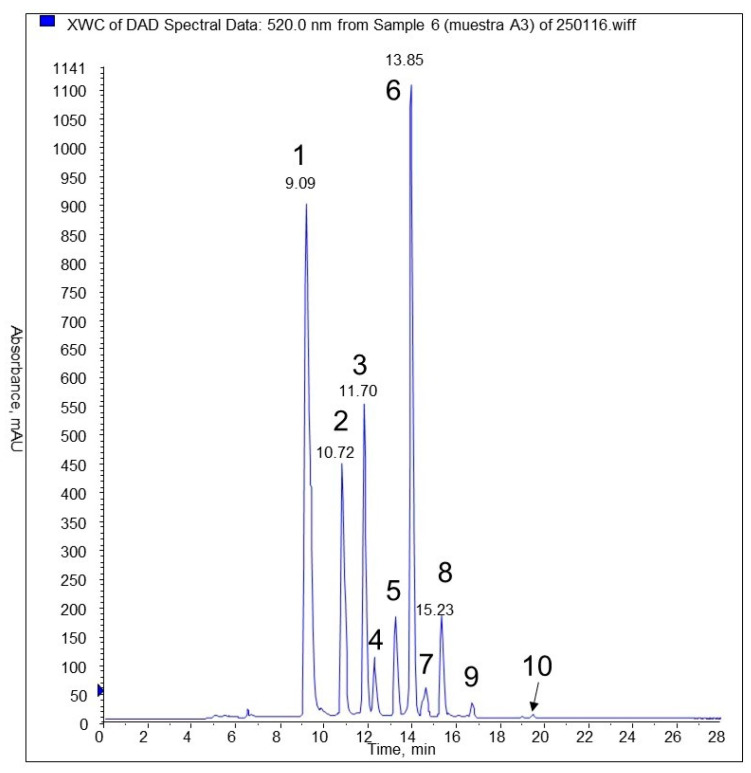
Chromatogram of *E. rubrum* fruit extract F3. Identification numbers in Table 5.

**Figure 4 plants-13-00497-f004:**
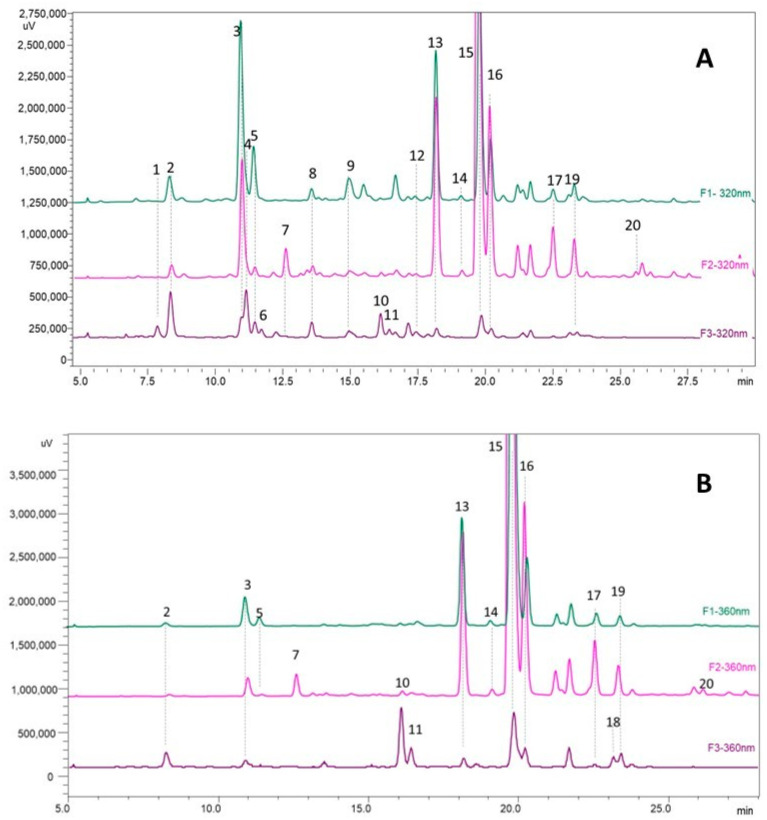
Chromatograms of F1, F2, and F3 extracts at 320 nm for hydroxycinnamic acid derivates (**A**) and 360 nm for flavonol determination (**B**). Identification in Table 6.

**Figure 5 plants-13-00497-f005:**
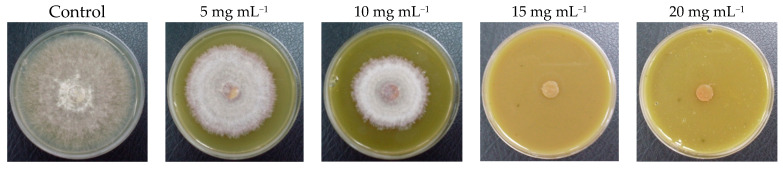
Inhibition of mycelium growth with increasing MEE concentrations (5 mg mL^−1^ to 20 mg mL^−1^) against *R. solani* after 84 h of cultivation.

**Figure 6 plants-13-00497-f006:**
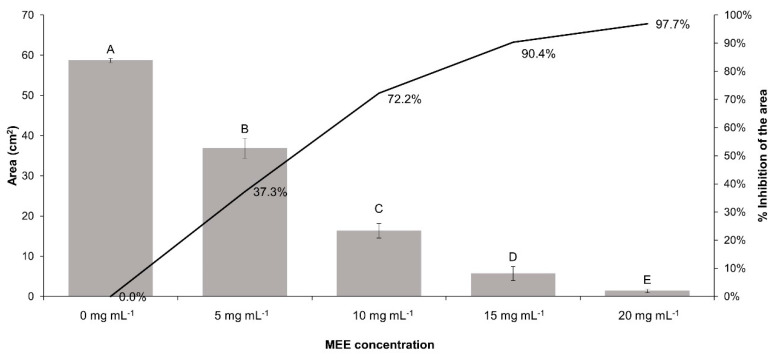
Influence on mycelial growth according to increasing concentrations of MEE. The different letters indicate statistically significant differences between the MEE concentrations under the Tukey test results (*p* ≤ 0.05).

**Figure 7 plants-13-00497-f007:**
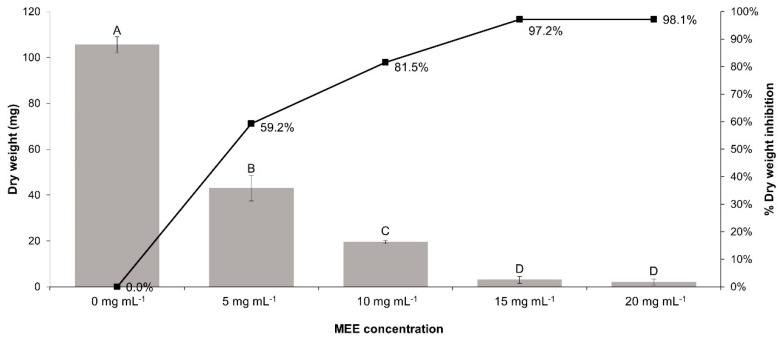
Relation between MEE concentrations and the accumulated biomass of *R. solani*. There are no statistically significant differences between the same letters, based on the results of the Tukey test (*p* ≤ 0.05).

**Table 1 plants-13-00497-t001:** Antioxidant activity of MEE and MEF expressed in IC_50_.

Extract	Radical	Replicates	Polynomial Equation	IC_50_	IC_50_ (av. ± SD)
MEE	DPPH	R_1_	y = −28.847x^2^ + 137.19x − 1.1257	0.4076 mg mL^−1^	0.4145 ± 0.0068 mg mL^−1^
		R_2_	y = −28.818x^2^ + 134.77x − 0.928	0.4146 mg mL^−1^	
		R_3_	y = −28.269x^2^ + 134.84x − 1.7967	0.4213 mg mL^−1^	
	ABTS	R_1_	y = −14.472x^2^ + 445.41x + 0.5478	0.1114 mg mL^−1^	0.1088 ± 0.0023 mg mL^−1^
		R_2_	y = −166.71x^2^ + 476.19x + 1.035	0.1068 mg mL^−1^	
		R_3_	y = −52.636x^2^ + 463.07x + 0.5131	0.1082 mg mL^−1^	
MEF	DPPH	R_1_	y = 0.3167x^2^ + 3.5126x + 13.915	6.4833 mg mL^−1^	6.4768 ± 0.0218 mg mL^−1^
		R_2_	y = 0.3805x^2^ + 2.7122x + 16.336	6.4946 mg mL^−1^	
		R_3_	y = 0.2452x^2^ + 4.2858x + 12.138	6.4524 mg mL^−1^	
	ABTS	R_1_	y = −15.796x^2^ + 79.533x − 2.1971	0.7758 mg mL^−1^	0.7666 ± 0.0089 mg mL^−1^
		R_2_	y = −15.654x^2^ + 78.764x − 0.7149	0.7581 mg mL^−1^	
		R_3_	y = −15.885x^2^ + 79.719x − 1.7466	0.7660 mg mL^−1^	

**Table 2 plants-13-00497-t002:** Antioxidant activity of MEE expressed in GAE and TEAC.

Radical	GAE *	TEAC **
DPPH	26.31 ± 1.51 mg	-
ABTS	-	200.24 ± 2.61 mg

* The expression of the result is mg of gallic per 1000 mg (1 g) of MEE. ** The expression of the result is mg of Trolox per 1000 mg (1 g) of MEE.

**Table 3 plants-13-00497-t003:** Antioxidant activity of MEF expressed in GAE and TEAC.

Radical	GAE *	TEAC **
DPPH	1.77 ± 0.24 mg	-
ABTS	-	53.22 ± 2.06 mg

* The expression of the result is mg of gallic per 1000 mg (1 g) of MEF. ** The expression of the result is mg of Trolox per 1000 mg (1 g) of MEF.

**Table 4 plants-13-00497-t004:** Results of preliminary qualitative assays for groups of secondary metabolites.

Secondary Metabolites	Leaves and Stems	MEE
Flavonoids	+++	+++
Hydrolizable Tannins	−	−
Condensed Tannins	+++	+++
Coumarins	−	−
Saponins	−	−
Alkaloids	−	−

Marked presence of metabolites (+++), and (−) indicates the absence of secondary metabolites.

**Table 5 plants-13-00497-t005:** Identification of main anthocyanins present in *E. rubrum* fruit methanolic extract by HPLC-DAD-ESI-MS/MS.

Identification	N° Peak	tr (min)	λ (nm)	[M + H]^+^	Fragments
delphinidin-3-glucoside (*)	1	9.09	523	465	303.9
cyanidin-3-glucoside (*)	2	10.7	520	449	287
petunidin-3-glucoside (*)	3	11.7	525	479	317
cyanidin pentoside	4	12.14	519	419	287
petunidin pentoside	5	13.14	522	449	317; 302; 274
peonidina-3-glucoside (*)	5	13.14	522	463	301; 286
malvidin-3-glucoside (*)	6	13.85	527	493	331; 315; 287
peonidin pentoside	7	14.49	522	433	301; 286; 158
malvidin pentoside	8	15.24	528	463	331; 315; 287
malvidin derivate	9	16.6	527	521	331
malvidin	10	19.38	535	331	

Note: (*): identification confirmed with commercial standards.

**Table 6 plants-13-00497-t006:** Identification of the main phenolic compounds present in *E. rubrum* extracts by HPLC-DAD-ESI-MS/MS.

N° Peak	Identifications	t_R_ (min)	DAD (nm)	[M − H]^−^	Fragments
1	coumaric acid derivate	7.84	320	361	163, 119
2	caffeoylquinic acid isomer	8.33	324	535	191
3	caffeoylquinic acid isomer	10.9	324	553	191
4	caffeoylquinic acid isomer	11.11	324	553	191
5	coumaroylquinic acid	11.43	306	337	191, 163, 119, 155
6	coumaric acid derivate	11.65	284	325	163, 119
7	myricetin-3-galactoside (*)	12.5	340	479	317, 287, 271
8	feruloylquinic acid	13.52	320	367	161, 133
9	coumaroylquinic acid	14.9	320	337	191, 173
10	myricetin-3-glucoside (*)	16.05	360	479	317, 287, 271
11	myricetin rutinoside	16.1	360	625; 479	317, 287, 271
12	quercetin-3-rutinoside (*)	17.93	360	609	301
13	caffeic acid derivate	18.1	320	367	179, 191, 161, 135
14	quercetin galactoside	19.33	360	463	301
15	laricitin hexoside	19.75	360	493	331, 315, 287
16	quercetin-3-glucoside (*)	20.12	360	463	301, 271, 179, 163
17	quercetin pentoside	22.35	355	433	301, 271, 255, 151
18	kaempferol-3-glucoside (*)	23.1	348	447	285, 255, 227, 151
19	isorhamnetin hexoside	23.67	351	477	315, 285, 299, 271
20	unknown quercetin hexoside	26.8	354	583	463, 301

Note: (*): identification confirmed with commercial standards.

## Data Availability

The data presented in this manuscript are available from the authors upon reasonable request.

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
