# Peer review of "Antioxidant and Antifungal Activities and Characterization of Phenolic Compounds Using High-Performance Liquid Chromatography and Mass Spectrometry (HPLC-MS) in Empetrum rubrum Vahl ex Willd."

_plants, 2024, doi:10.3390/plants13040497_

Round 1

Reviewer 1 Report

Comments and Suggestions for Authors

The manuscript entitled “ Antioxidant and antifungal activities and characterization of 2 phenolic compounds using HPLC- MS in Empetrum rubrum 3 Vahl ex Willd” is well designed and contains several assays that improved the total work. However, Some issues should be resolved prior to publication:

-The abstract:

You should first follow the steps in writing an abstract, the methods should be written first (all) the  next paragraph should be the results.

-The introduction

Line 64 to the end, please provide one name of the plant ( not to use the synonyms) it is confusing and to the reader as you talk about another plant.

-Methods

In the Evaluation of the Inhibitory Activity in Relation to the Fungal Growth, please provide with  pictures of the dishes.

Author Response

Dear Dr.:

After considering the Comments and Suggestions for Authors, I am submitting the new Manuscript titled “Antioxidant and antifungal activities and characterization of phenolic compounds using HPLC- MS in Empetrum rubrum Vahl ex Willd.”. In the new manuscript I have included a new author:

Name: Camila Fuica

E mail: camilafuica@gmail.com

Best regards

Dr. Carlos Schneider

Reviewer 2 Report

Comments and Suggestions for Authors

The article titled: Antioxidant and antifungal activities and characterization of phenolic compounds using HPLC- MS in Empetrum rubrum Vahl ex Willd. presents an interesting study. However, the article needs to be completed/ or verified. Below, I give my comments. 

1. The abbreviations should be explained – MME is explained only in the Abstract. The Abstract is a separate part of the manuscript; therefore, the abbreviations that were explained in the Abstract should be explained once again.

2. I suggest improving keywords. Different words compared with the title will enhance the visibility of the manuscript on the net.

3. Extraction method: Which volume of solvent was used to extract the plant, and how much weight of the plant was used (g)? The ratio of solvent to the plant is useful. 

4. I understood that the extraction was carried out for numerous hours at 65°C. Why? Is it justified? Are the Authors sure that the compounds included in the extracts are not degraded? If it is – the results of biological activity are not true. 

5. To clearly visualize the level of antioxidant effect, I suggest that the authors also include the IC50 of the reference substance. It will additionally help assess the potency of the tested extract.

6. Extracts from fruits should also be biologically tested. The authors presented only the data above the phytochemical content of fruit extract; thus, the biological potential should also be studied.

7. Tables should be prepared using the same font as in the main manuscript

8. The references are mixed, e.g., 42 and 43 are rather 41 and 42 …. ?

9. Why was R. solani chosen to test? Is there some connection between the antioxidant direction of activity and this antimicrobial action? More bacterial strains should be tested - Authors conclude the antimicrobial activity.

10. The conclusion should be more refined and precise. Moreover, the statement that phenol compounds in E. rubrum could have an antifungal effect against R. solani is not well related to the research: The authors did not study the compounds but the extracts. 

Author Response

(The authors gave the same response as above.)

Reviewer 3 Report

Comments and Suggestions for Authors

The author Carlos Schneider reported manuscript with title Antioxidant and antifungal activities and characterization of 2 phenolic compounds using HPLC- MS in Empetrum rubrum 3 Vahl ex Willd. The topic is very interesting and the experimental design is good therfore i would like to recommend for publication after minor changes.

1. revised the bastract with main results highlight

2. explain the role of biomolecules in plant as antioxidant compounds in introduction.

3. Research gap mention, why this study is important.

4. methodology need to revise specially units 

5. figures need to be high resolution 

6. references should be updates and cite the following paper.

  • DOI: 
  • 10.3390/biomedicines11102832
  • DOI: 
  • 10.1016/j.apsadv.2023.100446
Comments on the Quality of English Language

Minor changes 

Author Response

(The authors gave the same response as above.)

Round 2

Reviewer 2 Report

Comments and Suggestions for Authors

Thank you to the authors for the amendments introduced to the manuscript. However, I still see the following small inaccuracies:

1. There is not a conclusion in the abstract.

2. There are the values of approximation of antioxidant activity in the abstract (DPPH and ABTS methods with MEE were 0.4145 mg mL-1 and 0.1081 mg mL), different from the main text (0.414 and 0.108). 

3. The IC50 was presented without the standard deviation. Why? It should be completed.

Author Response

Reviewer 2 (Round 2)

Dear Dr.,

I am informing you about the corrections that I have made in the manuscript “plants-2752809 (Minor Revisions were corrected)”. I am sending “plants-2752809 (Minor Revisions were corrected)” as an attachment.

  1. There is not a conclusion in the abstract.

An abbreviated conclusion was added in the abstract, in lines 27 and 28 of “plants-2752809 (Minor Revisions were corrected)”

  1. There are the values of approximation of antioxidant activity in the abstract (DPPH and ABTS methods with MEE were 0.4145 mg mL-1 and 0.1081 mg mL), different from the main text (0.414 and 0.108). 

According to the requested correction, the IC50 values in the Abstract are the same as those in Table 1, in the Results of “plants-2752809 (Minor Revisions were corrected)”.

  1. The IC50 was presented without the standard deviation. Why? It should be completed.

According to the dates of Table 1 of “plants-2752809 (Minor Revisions were corrected)”, the standard deviation of IC50 values was calculated and added.

Best regards

Dr. Carlos Schneider
